# Design, Synthesis, and Anticancer Activity Studies of Novel Quinoline-Chalcone Derivatives

**DOI:** 10.3390/molecules26164899

**Published:** 2021-08-13

**Authors:** Yong-Feng Guan, Xiu-Juan Liu, Xin-Ying Yuan, Wen-Bo Liu, Yin-Ru Li, Guang-Xi Yu, Xin-Yi Tian, Yan-Bing Zhang, Jian Song, Wen Li, Sai-Yang Zhang

**Affiliations:** 1School of Chemical Engineering, Zhengzhou University, Zhengzhou 450001, China; gyf169@126.com; 2Key Laboratory of Advanced Drug Preparation Technologies (Ministry of Education), Institute of Drug Discovery & Development, School of Pharmaceutical Sciences, Zhengzhou University, Zhengzhou 450001, China; LXJ120WYY@163.com (X.-J.L.); yxy17865562065@163.com (X.-Y.Y.); 17698567571@163.com (W.-B.L.); zhangyb@zzu.edu.cn (Y.-B.Z.); 3School of Basic Medical Sciences, Zhengzhou University, Zhengzhou 450001, China; 17603868158@163.com (Y.-R.L.); ygx19990110@126.com (G.-X.Y.); txyi626@163.com (X.-Y.T.)

**Keywords:** quinoline, chalcone, anticancer activity, cell cycle arrest, ROS

## Abstract

The chalcone and quinoline scaffolds are frequently utilized to design novel anticancer agents. As the continuation of our work on effective anticancer agents, we assumed that linking chalcone fragment to the quinoline scaffold through the principle of molecular hybridization strategy could produce novel compounds with potential anticancer activity. Therefore, quinoline-chalcone derivatives were designed and synthesized, and we explored their antiproliferative activity against MGC-803, HCT-116, and MCF-7 cells. Among these compounds, compound **12e** exhibited a most excellent inhibitory potency against MGC-803, HCT-116, and MCF-7 cells with IC_50_ values of 1.38, 5.34, and 5.21 µM, respectively. The structure–activity relationship of quinoline-chalcone derivatives was preliminarily explored in this report. Further mechanism studies suggested that compound **12e** inhibited MGC-803 cells in a dose-dependent manner and the cell colony formation activity of MGC-803 cells, arrested MGC-803 cells at the G2/M phase and significantly upregulated the levels of apoptosis-related proteins (Caspase3/9 and cleaved-PARP) in MGC-803 cells. In addition, compound **12e** could significantly induce ROS generation, and was dependent on ROS production to exert inhibitory effects on gastric cancer cells. Taken together, all the results suggested that directly linking chalcone fragment to the quinoline scaffold could produce novel anticancer molecules, and compound **12e** might be a valuable lead compound for the development of anticancer agents.

## 1. Introduction

Chalcone is a natural product template which shows many versatile pharmacological activities especially anticancer activities [1,2,3]. Due to its simple chemistry and ease of synthesis, a large number of chalcone derivatives was discovered with variety of promising biological activity [4]. In fact, chalcone compounds have shown good therapeutic effects and clinical application potential as anticancer drugs for the treatment of human cancers [5,6,7]. In addition, chalcone fragment was also frequently utilized to design novel agents with other anticancer moieties to enhance the biological efficacy by the molecular hybridization strategy [8,9,10,11,12,13,14,15]. 4-Aminochalcone derivative **1** [12] displayed excellent inhibitory activity against NCI-H460, A549, and H1975 cells with IC_50_ values of 2.3, 3.2, and 5.7 μM, respectively. Compound **1** was able to trigger ROS-mediated apoptosis in time- and concentration-dependent manners in NCI-H460 cells. In addition, compound **1** also displayed a better safety profile in animal models. Chalcone dithiocarbamate derivative **2** [13] was reported as a LSD1 inhibitor with an IC_50_ value of 0.14 μM. Compound **2** exhibited potent anticancer activity against MOLT-4 cells (IC_50_ = 0.87 μM) and was significantly effective in suppressing the growth of MOLT-4 xenograft tumor mouse model. Compound **3** [14] was successfully designed by the structural combination of the 1,3,5-triazine and chalcone fragments via a diether linker with potent antiproliferative activity against the MCF-7 and HCT116 cells (GI_50_ = 0.127 and 0.116 μM, respectively). Additionally, compound **3** could potently inhibit the activity of cellular DHFR and TrxR in HCT-116 cells. [1,2,4] Triazolo [1,5-*a*] pyrimidine–chalcone derivative **4** [15] inhibited MGC-803 cells at the nanomolar level (GI_50_ = 0.64 μM) and exhibited its anti-proliferative potency via inducing autophagy and increasing ROS level (Figure 1). Therefore, chalcone fragment might be a valuable and core moiety to design anticancer agents. In this work, we continued our efforts on the discovery of chalcone derivatives as potential anticancer activity.

Quinolines, as one class of *N*-containing heterocycles, have numerous advantages over other non-nitrogenous, which are widely used as “parental” compounds to synthesize molecules with variety of promising biological activity [16,17,18,19]. The quinoline motifs are frequently found in many compounds that show potent anticancer activity with different mechanisms [20,21,22,23,24,25,26,27]. Anlotinib [23] (multi-kinase inhibitor) and Bosutinib [24] (Src-Abl inhibitor), which are quinoline-based protein kinase inhibitors, have been approved for the treatment of human cancers. Quinoline–chalcone derivative **5** [25] as a potent tubulin inhibitor showed excellent anticancer potency with IC_50_ values at nanomolar levels. In addition, compound **5** could arrest the cell cycle at the G2/M phase, induce apoptosis, depolarize mitochondria, and induce ROS generation in K562 cells. Quinoline chalcone **6** [26] is effective in exhibiting potent activity against HL60 cells with an IC_50_ values of 0.59 μM. Novel phenylsulfonylurea derivative **7** [27] as an anticancer agent exhibited potent cytotoxicity activity against HepG-2, A549, and MCF-7 cells (IC_50_ = 2.71, 7.47, and 6.55 μM, respectively) as well as moderate PI3K/mTOR dual inhibitory activity (Figure 2). Therefore, in this work, we use the quinoline moiety as the core scaffold of molecules to discover novel quinoline-based anticancer agents.

Molecular hybridization strategy is extensively used in drug design and discovery based on the combination of different bioactive moieties to produce new hybrids with the improved activities [28]. These interesting findings about chalcones and quinolines as anticancer agents led to molecular hybridization strategy of chalcone and quinoline scaffolds to generate novel anticancer agents. In this work, as the continuation of our work on the development of anticancer agents, we designed and synthesized novel quinoline-chalcone derivatives as anticancer activity (Figure 3).

## 2. Results and Discussion

### 2.1. Chemistry

Target quinoline-chalcone derivatives were synthesized by outlined procedures in Scheme 1. Commercially available 4-aminoacetophenone (**8**) reacted with aromatic aldehydes **9a**–**9j** to afford compounds **10a**–**10j** in the presence of NaOH in EtOH at 25 °C. Substitution reaction between compounds **10a**–**10j** with commercially available 4-chloro-2-methylquinoline (**11**) gave target compounds **12a**–**12j** in the presence of HCl in EtOH at 80 °C. In addition, Compounds **12a**, **12b**, **12e**, and **12f** then reacted with iodomethane or iodoethane in the presence of KOH in acetonitrile at 80 °C to give compounds **13a**–**13f**. All the compounds were characterized by means of NMR and HREI-mass spectra which are showed in the Appendix A.

### 2.2. Antiproliferative Activity and Structure Activity Relationship Analysis

According to the latest statistics from the International Agency for Research on Cancer (IARC), in 2020, the number of newly diagnosed patients of colorectal cancer, gastric cancer, and breast cancer ranked second, third, and fourth in China, respectively [29]. Therefore, the in vitro antiproliferative activities of new target compounds **12a**–**12j** and **13a**–**13f** were evaluated against MGC-803 cell line (human gastric cancer cells), HCT-116 cell line (human colon cancer cells), and MCF-7 (human breast cancer cells), with the 5-fluorouracil (5-Fu**)** as a positive control. The following Table 1 depicted the results of antiproliferative activity.

As shown in Table 1, most of quinoline-chalcone derivatives displayed potent antiproliferative activity against MGC-803, HCT-116 and MCF-7 cells with IC_50_ values <20 µM. Particularly, compound **12e** exhibited most excellent inhibitory potency against MGC-803, HCT-116, and MCF-7 cells with IC_50_ values of 1.38, 5.34, and 5.21 µM, respectively, which were much lower than that of 5-Fu (IC_50_ values = 6.22 µM, 10.4 μM, and 11.1 μM, respectively), which indicated that compound **12e** was effective in inhibiting the activity of three kinds of tumor cells. In addition, most of compounds was more sensitive to MGC-803 cells than that of HCT-116 and MCF-7 cells. Therefore, the structure–activity relationships were discussed according to the results of antiproliferative activity of MGC-803 cells. As shown in Table 1, the types and positions of substituents (R_1_) on chalcone group (A ring) have an important influence on its antiproliferative activity. Compared with **12f**, compounds **12a**–**12e** with electron-donating groups of A ring exhibited enhanced activity than compounds without substitution groups of A ring, but compounds **12g** and **12i**, with electron-withdrawing groups of A ring, despaired the antiproliferative activity more potent than that of compound **12f**. In addition, the position of substituents (R_1_) is also important. The inhibitory activity of compounds was less potent when the substituents (R_1_) were at the 3-position of chalcone group (A ring) than that of the substituents (R_1_) were at the 3-position of A ring (compounds **12b** vs. **12c**, **12g** vs. **12i**, and **12h** vs. **12j**). However, compound **12e** with a 3,4,5-triOCH_3_ substituent of chalcone group (A ring) exhibited better activity. The relationships between the electron-donating groups and electron-withdrawing groups of chalcone group (A ring) and the inhibitory potency on MGC-803 cells were 3,4,5-triOCH_3_ > 3,4-diOCH_3_ > 4-CH_3_ > 4-Br > 4-OCH_3_ > 3-OCH_3_ > 3-Br > H > 4-Cl > 3-Cl. Next, the influence of R_2_ was further explored. As shown in Table 1, the inhibitory activity of compounds **13a**–**13f** was decreased when the H group was replaced by CH_3_ or CH_3_CH_2_ substituent (compounds **13a** vs. **12f**, **13b** vs. **12a**, **13c** vs. **12d**, **13d** vs. **12g**, **13e** vs. **12e**, and **13f** vs. **12e**), indicating that the substituents of R_2_ could not improve the inhibitory potency.

Based on above the antiproliferative activity results of compounds, we can conclude that linking quinoline fragment to the chalcone scaffold produces new hybrids with potential anticancer activity. The types and positions of substituents (R_1_) on chalcone group (A ring) make a great influence on the inhibitory potency of compounds. Substituents of R_2_ impaired the inhibitory potency of compounds (Figure 4).

### 2.3. Compound ***12e*** Inhibited Gastric Cancer Cells

Compound **12e** in this series of compounds showed the most excellent inhibitory activity against MGC-803 cells. The cell viability of the MGC-803 cells was significantly decreased after the treatment with different concentrations of compound **12e** for 48 h (Figure 5D). The cell inhibition rate of the high concentration treatment group increased by more than 60%. Cell proliferation inhibition could be caused by cell cycle arrestment. As shown in Figure 5A,C, the results of cell cycle analysis showed that gastric cancer cells MGC-803 could arrest the cells in G2/M phase. The percentage of cells was upregulated by 20%. Compound **12e** also showed significant activity on inhibiting the activity of cell colony formation (Figure 5B). With long-term low concentration treatment of compound **12e**, the formatted colony was significantly decreased from the concentration of 900 nM. The detection of apoptosis-related proteins showed that the levels of cleaved-Caspase3/9 and cleaved-PARP, the markers of apoptosis, were significantly upregulated (Figure 5E). In conclusion, compound **12e** could inhibit gastric cancer cells by arresting cells in G2/M phase and inducing apoptosis of MGC-803 cells.

### 2.4. Compound ***12e*** Inhibited Gastric Cancer Cells through the Generation of Reactive Oxygen Species (ROS)

It has been reported in the literature that the molecular skeleton of chalcone can induce the production of ROS [12,15]. The assay labeling ROS using DCFH-DA probe revealed that compound **12e** could effectively induce ROS generation at concentrations as low as 500 nM (Figure 6A). To investigate the direct relationship between the induction of ROS generation and tumor cell inhibition by compound **12e**, gastric cancer cells (MGC-803 and SGC-7901 cells) were treated with the ROS inhibitor NAC in combination with compound **12e**. As shown in Figure 6B,C, the results showed that the presence of NAC significantly reversed the cytostatic effect caused by compound **12e**. The above results suggested that compound **12e** could significantly induce ROS generation and was dependent on ROS production to exert inhibitory effects on gastric cancer cells.

## 3. Materials and Methods

All the chemical reagents were purchased from commercial suppliers (Energy chemical Company, Shanghai, China). Melting points were determined on an X-5 micromelting apparatus (Fukai Instrument, Beijing, China). NMR spectra data was recorded with a Bruker DPX 400 MHz spectrometer (Bruker, Billerica, MA, USA). High-resolution mass spectra (HRMS) data was obtained using a Waters Micromass Q-TOF Micromass spectrometer (Waters, Manchester, UK) by electrospray ionization (ESI).

### 3.1. Synthesis of Compounds ***10a**–**10j***

A solution of commercially available 4-aminoacetophenone **8** (1.0 mmol, 1.0 eq), aromatic aldehyde **9a**–**9j** (1.0 mmol, 1.0 eq), NaOH (2.0 mmol, 2.0 eq) and were added into 20 mL EtOH at 25 °C. After 6 h, 20 mL water was added to the reaction mixture, giving yellow solid. Then, the crude product was obtained by filtration without further purification.

### 3.2. Synthesis of Compounds ***12a**–**12j*** and ***13a**–**13f***

A solution of compounds **10a**–**10j** (1.0 mmol, 1.0 eq), 4-chloro-2-methylquinoline **11** (1.0 mmol, 1.0 eq) were added into 10 mL EtOH at 80 °C in the presence of HCl. After 8 h, organic phases were evaporated to obtain crude products, and then were purified to give compounds **12a**–**12j** by column chromatography. A solution of compounds **12a**, **12b**, **12e**, **12f**, and **12e** (1.0 mmol, 1.0 eq), iodomethane or iodoethane (2.0 mmol, 2.0 eq), and KOH (2.0 mmol, 2.0 eq) were added into 10 mL acetonitrile at 80 °C. After 6 h, the reaction was quenched with water and the crude product extracted with ethyl acetate three times, organic phases were evaporated to obtain crude products, then were purified to give compounds **13a**–**13f** by column chromatography (PE:EA = 5:1).

(E)-1-(4-((2-methylquinolin-4-yl)amino)phenyl)-3-(p-tolyl)prop-2-en-1-one (**12a**). Yellow powder, Yield, 68%, m.p.124–126 °C. ^1^H NMR (400 MHz, DMSO-d_6_) δ 9.31 (s, 1H), 8.30 (dd, J = 8.5, 1.4 Hz, 1H), 8.24–8.18 (m, 2H), 7.94 (d, J = 15.6 Hz, 1H), 7.87 (d, J = 8.3 Hz, 1H), 7.82–7.76 (m, 2H), 7.73 (d, J = 7.3 Hz, 1H), 7.71–7.67 (m, 1H), 7.56–7.51 (m, 1H), 7.50 (s, 1H), 7.48 (s, 1H), 7.33–7.23 (m, 3H), 2.56 (s, 3H), 2.36 (s, 3H). ^13^C NMR (100 MHz, DMSO-d_6_) δ 187.52, 159.42, 149.29, 147.00, 146.09, 143.55, 140.90, 132.64, 131.52, 130.93, 130.01, 129.96, 129.26, 129.08, 124.85, 122.68, 121.47, 119.93, 119.06, 106.02, 25.60, 21.55. HRMS m/z calcd. for C_26_H_23_N_2_O, [M + H]^+^: 379.1805, found: 379.1812.

*(E)-3-(4-methoxyphenyl)-1-(4-((2-methylquinolin-4-yl)amino)phenyl)prop-2-en-1-one* (**12b**). Yellow powder, Yield, 64%, m.p.111–113 °C. ^1^H NMR (400 MHz, DMSO-*d*_6_) δ 9.29 (s, 1H), 8.29 (dd, *J* = 8.5, 1.4 Hz, 1H), 8.24–8.16 (m, 2H), 7.89–7.82 (m, 4H), 7.74–7.66 (m, 2H), 7.50 (dd, *J* = 10.1, 7.1 Hz, 3H), 7.24 (s, 1H), 7.07–6.99 (m, 2H), 3.83 (s, 3H), 2.55 (d, *J* = 2.3 Hz, 3H). ^13^C NMR (100 MHz, DMSO-*d*_6_) δ 187.46, 159.42, 149.38, 146.23, 144.70, 138.42, 131.92, 130.65, 129.96, 129.05, 128.04, 124.84, 122.69, 119.90, 119.56, 119.24, 119.17, 113.09, 110.95, 105.81, 55.95, 25.57. HRMS *m*/*z* calcd. for C_26_H_23_N_2_O_2_, [M + H]^+^: 395.1754, found: 395.1758.

*(E)-3-(3-methoxyphenyl)-1-(4-((2-methylquinolin-4-yl)amino)phenyl)prop-2-en-1-one* (**12c**). Yellow powder, Yield, 66%, m.p.173–175 °C. ^1^H NMR (400 MHz, DMSO-*d*_6_) δ 9.32 (s, 1H), 8.29 (dd, *J* = 8.5, 1.4 Hz, 1H), 8.26–8.19 (m, 2H), 8.00 (d, *J* = 15.6 Hz, 1H), 7.87 (dd, *J* = 8.5, 1.3 Hz, 1H), 7.75–7.68 (m, 2H), 7.57–7.48 (m, 4H), 7.44 (dt, *J* = 7.8, 1.4 Hz, 1H), 7.38 (t, *J* = 7.8 Hz, 1H), 7.26 (s, 1H), 7.03 (ddd, *J* = 8.1, 2.6, 1.0 Hz, 1H), 3.85 (s, 3H), 2.56 (s, 3H). ^13^C NMR (100 MHz, DMSO-*d*_6_) δ 193.40, 160.16, 159.44, 143.53, 137.15, 137.08, 131.12, 131.06, 130.37, 130.04, 129.86, 128.97, 128.70, 124.94, 122.68, 122.03, 119.97, 119.08, 118.93, 114.81, 114.69, 113.64, 55.42, 25.49. HRMS *m*/*z* calcd. for C_26_H_23_N_2_O_2_, [M + H]^+^: 395.1754, found: 395.1760.

*(E)-3-(3,4-dimethoxyphenyl)-1-(4-((2-methylquinolin-4-yl)amino)phenyl)prop-2-en-1-one* (**12d**). Yellow powder, Yield, 69%, m.p.153–155 °C. ^1^H NMR (400 MHz, DMSO-*d*_6_) δ 9.30 (s, 1H), 8.30 (dd, *J* = 8.5, 1.4 Hz, 1H), 8.25–8.18 (m, 2H), 7.92–7.83 (m, 2H), 7.75–7.66 (m, 2H), 7.57 (d, *J* = 2.0 Hz, 1H), 7.54–7.46 (m, 3H), 7.39 (dd, *J* = 8.4, 2.0 Hz, 1H), 7.25 (s, 1H), 7.03 (d, *J* = 8.4 Hz, 1H), 3.88 (s, 3H), 3.83 (s, 3H), 2.55 (s, 3H).^13^C NMR (100 MHz, DMSO-*d*_6_) δ 187.40, 159.40, 151.61, 149.53, 149.26, 146.84, 146.16, 144.02, 131.73, 130.88, 129.97, 129.08, 128.20, 124.84, 124.32, 122.68, 120.02, 119.90, 119.11, 112.04, 111.05, 105.87, 56.22, 56.07, 25.59. HRMS *m*/*z* calcd. for C_27_H_25_N_2_O_3_, [M + H]^+^: 425.1860, found: 425.1869.

*(E)-1-(4-((2-methylquinolin-4-yl)amino)phenyl)-3-(3,4,5-trimethoxyphenyl)prop-2-en-1-one* (**12e**). Brown power, Yield, 60%, m.p.122–124 °C. ^1^H NMR (400 MHz, DMSO-*d*_6_) δ 9.36 (s, 1H), 8.30 (dd, *J* = 8.5, 1.4 Hz, 1H), 8.26–8.20 (m, 2H), 7.95 (d, *J* = 15.5 Hz, 1H), 7.86 (d, *J* = 8.3 Hz, 1H), 7.75–7.65 (m, 2H), 7.54–7.48 (m, 3H), 7.25 (d, *J* = 3.5 Hz, 3H), 3.88 (s, 6H), 3.73 (s, 3H), 2.55 (s, 3H). ^13^C NMR (100 MHz, DMSO-*d*_6_) δ 187.46, 153.60, 152.85, 144.00, 140.09, 131.49, 131.01, 130.91, 130.01, 129.03, 124.90, 122.68, 121.65, 119.92, 119.05, 106.89, 105.21, 60.63, 56.62, 56.19, 25.56. HRMS *m*/*z* calcd. for C_28_H_27_N_2_O_4_, [M + H]^+^: 455.1965, found: 455.1975.

*(E)-1-(4-((2-methylquinolin-4-yl)amino)phenyl)-3-phenylprop-2-en-1-one* (**12f**). Yellow powder, Yield, 58%, m.p.265-267 °C. ^1^H NMR (400 MHz, DMSO-*d*_6_) δ 11.10 (s, 1H), 8.89 (d, *J* = 8.5 Hz, 1H), 8.34 (d, *J* = 8.6 Hz, 2H), 8.17 (d, *J* = 8.4 Hz, 1H), 8.05–7.98 (m, 2H), 7.92 (dd, *J* = 6.7, 3.0 Hz, 2H), 7.83–7.76 (m, 2H), 7.74 (dd, *J* = 6.7, 4.8 Hz, 2H), 7.52–7.44 (m, 3H), 7.04 (s, 1H), 2.69 (s, 3H). ^13^C NMR (100 MHz, DMSO-*d*_6_) δ 192.85, 160.84, 153.04, 151.68, 149.78, 136.85, 135.87, 131.12, 130.30, 129.59, 129.45, 128.90, 128.72, 128.54, 127.31, 126.55, 123.61, 123.54, 120.28, 114.25, 25.33. HRMS *m*/*z* calcd. for C_25_H_21_N_2_O, [M + H]^+^: 365.1648, found: 365.1655.

*(E)-3-(4-chlorophenyl)-1-(4-((2-methylquinolin-4-yl)amino)phenyl)prop-2-en-1-one* (**12g**). Yellow powder, Yield, 58%, m.p.203–205 °C. ^1^H NMR (400 MHz, DMSO-*d*_6_) δ 11.07 (s, 1H), 8.86 (d, *J* = 8.6 Hz, 1H), 8.34 (d, *J* = 8.2 Hz, 2H), 8.15 (d, *J* = 8.5 Hz, 1H), 7.99 (dd, *J* = 22.2, 7.4 Hz, 4H), 7.79 (s, 1H), 7.74 (d, *J* = 6.3 Hz, 2H), 7.71 (s, 1H), 7.54 (d, *J* = 8.2 Hz, 2H), 7.04 (s, 1H), 2.68 (s, 3H). ^13^C NMR (100 MHz, DMSO-*d*_6_) δ 188.29, 155.64, 153.94, 143.14, 142.67, 138.98, 135.67, 135.61, 134.27, 134.14, 131.13, 130.85, 129.48, 127.25, 124.78, 124.20, 123.10, 120.26, 117.24, 101.84, 20.32. HRMS *m*/*z* calcd. for C_25_H_20_ClN_2_O, [M + H]^+^: 399.1259, found: 399.1261.

*(E)-3-(4-bromophenyl)-1-(4-((2-methylquinolin-4-yl)amino)phenyl)prop-2-en-1-one* (**12h**). Yellow powder, Yield, 67%, m.p.133–135 °C. ^1^H NMR (400 MHz, DMSO-*d*_6_) δ 9.32 (s, 1H), 8.29 (dd, *J* = 8.5, 1.4 Hz, 1H), 8.24–8.18 (m, 2H), 8.03 (d, *J* = 15.6 Hz, 1H), 7.87 (d, *J* = 8.5 Hz, 3H), 7.73–7.65 (m, 4H), 7.53 (d, *J* = 7.4 Hz, 1H), 7.51–7.47 (m, 2H), 7.26 (s, 1H), 2.56 (s, 3H). ^13^C NMR (100 MHz, DMSO-*d*_6_) δ 192.55, 187.43, 142.10, 136.81, 135.12, 134.69, 132.35, 131.71, 131.59, 131.15, 131.07, 130.26, 130.02, 129.03, 128.92, 124.93, 122.70, 122.34, 118.83, 106.50, 25.56. HRMS *m*/*z* calcd. for C_25_H_20_BrN_2_O, [M + H]^+^: 443.0754, found: 443.0756.

*(E)-3-(3-chlorophenyl)-1-(4-((2-methylquinolin-4-yl)amino)phenyl)prop-2-en-1-one* (**12i**). Yellow powder, Yield, 63%, m.p.212–214 °C. ^1^H NMR (400 MHz, DMSO-*d*_6_) δ 9.55 (s, 1H), 8.35 (dd, *J* = 16.5, 8.3 Hz, 1H), 8.26–8.06 (m, 3H), 7.86 (ddd, *J* = 11.1, 6.2, 2.6 Hz, 2H), 7.77–7.61 (m, 2H), 7.56–7.46 (m, 5H), 7.35 (t, *J* = 2.2 Hz, 1H), 7.23 (d, *J* = 23.6 Hz, 1H), 2.55 (d, *J* = 5.1 Hz, 3H). ^13^C NMR (100 MHz, DMSO-*d*_6_) δ 187.37, 159.37, 149.33, 147.46, 146.10, 141.69, 137.67, 134.28, 131.15, 131.11, 131.07, 130.39, 129.94, 129.02, 128.31, 128.27, 124.82, 124.17, 123.02, 120.06, 118.91, 106.32, 25.60. HRMS *m*/*z* calcd. for C_25_H_20_ClN_2_O, [M + H]^+^: 399.1259, found: 399.1266.

*(E)-3-(3-bromophenyl)-1-(4-((2-methylquinolin-4-yl)amino)phenyl)prop-2-en-1-one* (**12j**). Yellow powder, Yield, 64%, m.p.219–221 °C. ^1^H NMR (400 MHz, DMSO-*d*_6_) δ 9.33 (d, *J* = 9.0 Hz, 1H), 8.29–8.22 (m, 3H), 8.08 (d, *J* = 15.6 Hz, 1H), 7.87 (dd, *J* = 8.2, 3.8 Hz, 2H), 7.74–7.62 (m, 3H), 7.55–7.47 (m, 3H), 7.41 (q, *J* = 7.6 Hz, 2H), 7.27 (s, 1H), 2.55 (d, *J* = 4.8 Hz, 3H). ^13^C NMR (100 MHz, DMSO-*d*_6_) δ 187.34, 159.43, 149.31, 147.30, 145.99, 141.69, 137.92, 133.30, 131.40, 131.17, 131.13, 129.98, 129.11, 128.68, 124.90, 124.10, 122.90, 122.69, 119.98, 118.91, 106.28, 25.61. HRMS *m*/*z* calcd. for C_25_H_20_BrN_2_O, [M + H]^+^: 443.0754, found: 443.0761.

*(E)-1-(4-(methyl(2-methylquinolin-4-yl)amino)phenyl)-3-phenylprop-2-en-1-one* (**13a**). Brown power, Yield, 57%, m.p.160–162 °C. ^1^H NMR (400 MHz, DMSO-*d*_6_) δ 8.07–7.99 (m, 3H), 7.92–7.82 (m, 3H), 7.73 (ddd, *J* = 8.4, 6.8, 1.5 Hz, 1H), 7.70–7.61 (m, 2H), 7.49–7.41 (m, 5H), 6.78–6.71 (m, 2H), 3.52 (s, 3H), 2.69 (s, 3H). ^13^C NMR (100 MHz, DMSO-*d*_6_) δ 188.47, 155.66, 153.97, 144.68, 142.52, 138.93, 135.74, 135.10, 134.31, 131.24, 130.82, 129.46, 129.41, 127.30, 124.81, 124.11, 122.33, 120.24, 117.19, 101.79, 25.92, 20.34. HRMS *m*/*z* calcd. for C_26_H_23_N_2_O, [M + H]^+^: 379.1805, found: 379.1809.

*(E)-1-(4-(methyl(2-methylquinolin-4-yl)amino)phenyl)-3-(p-tolyl)prop-2-en-1-one (***13b**). Yellow powder, Yield, 55%, m.p. 86–88 °C. ^1^H NMR (400 MHz, DMSO-*d*_6_) δ 8.06–7.99 (m, 3H), 7.83 (d, *J* = 15.6 Hz, 1H), 7.76–7.70 (m, 3H), 7.68–7.60 (m, 2H), 7.50–7.42 (m, 2H), 7.25 (d, *J* = 7.9 Hz, 2H), 6.78–6.71 (m, 2H), 3.52 (s, 3H), 2.69 (s, 3H), 2.34 (s, 3H). ^13^C NMR (100 MHz, DMSO-*d*_6_) δ 192.74, 160.79, 152.99, 151.74, 138.61, 137.05, 133.07, 131.08, 130.30, 129.99, 129.62, 129.54, 129.31, 127.51, 127.36, 126.51, 123.64, 123.52, 120.20, 114.33, 25.31, 21.52, 21.30. HRMS *m*/*z* calcd. for C_27_H_25_N_2_O, [M + H]^+^: 392.1889, found: 393.1964.

*(E)-3-(4-methoxyphenyl)-1-(4-(methyl(2-methylquinolin-4-yl)amino)phenyl)prop-2-en-1-one* (**13c**). Yellow powder, Yield, 63%, m.p.114–116 °C. ^1^H NMR (400 MHz, DMSO-*d*_6_) δ 8.05–7.98 (m, 3H), 7.80 (d, *J* = 8.8 Hz, 2H), 7.76–7.69 (m, 2H), 7.68–7.61 (m, 2H), 7.48–7.42 (m, 2H), 7.00 (d, *J* = 8.8 Hz, 2H), 6.78–6.70 (m, 2H), 3.81 (s, 3H), 3.52 (s, 3H), 2.68 (s, 3H). ^13^C NMR (100 MHz, DMSO-*d*_6_) δ 192.63, 187.44, 161.58, 160.94, 149.48, 143.10, 137.74, 131.60, 130.95, 130.84, 130.43, 129.28, 126.53, 123.74, 123.45, 119.91, 114.88, 114.68, 114.44, 114.06, 55.78, 55.60, 25.14. HRMS *m*/*z* calcd. for C_27_H_25_N_2_O_2_, [M + H]^+^: 409.1911, found: 409.1919.

*(E)-3-(4-chlorophenyl)-1-(4-(methyl(2-methylquinolin-4-yl)amino)phenyl)prop-2-en-1-one* (**13 d**). Yellow powder, Yield, 67%, m.p.159–161 °C. ^1^H NMR (400 MHz, DMSO-*d*_6_) δ 8.03 (dd, *J* = 8.2, 5.8 Hz, 3H), 7.94–7.87 (m, 3H), 7.73 (ddd, *J* = 8.4, 6.8, 1.5 Hz, 1H), 7.68–7.62 (m, 2H), 7.54–7.49 (m, 2H), 7.48–7.43 (m, 2H), 6.74 (d, *J* = 9.0 Hz, 2H), 3.53 (s, 3H), 2.69 (s, 3H). ^13^C NMR (100 MHz, DMSO-*d*_6_) δ 192.30, 160.83, 153.10, 151.63, 149.80, 135.96, 134.80, 133.47, 131.27, 131.15, 130.29, 129.62, 129.06, 128.74, 127.22, 126.53, 123.60, 123.53, 120.31, 114.22, 40.74. HRMS *m*/*z* calcd. for C_26_H_22_ClN_2_O, [M + H]^+^: 413.1415, found: 413.1420.

*(E)-1-(4-(methyl(2-methylquinolin-4-yl)amino)phenyl)-3-(3,4,5-trimethoxyphenyl)prop-2-en-1-one* (**13e**). Yellow powder, Yield, 61%, m.p.162–164 °C. ^1^H NMR (400 MHz, DMSO-*d*_6_) δ 8.10–7.98 (m, 3H), 7.84 (d, *J* = 15.5 Hz, 1H), 7.73 (ddd, *J* = 8.4, 6.8, 1.5 Hz, 1H), 7.67–7.58 (m, 2H), 7.51–7.43 (m, 2H), 7.18 (s, 2H), 6.79–6.71 (m, 2H), 3.85 (s, 6H), 3.71 (s, 3H), 3.53 (s, 3H), 2.69 (s, 3H). ^13^C NMR (100 MHz, DMSO-*d*_6_) δ 187.17, 160.78, 153.57, 152.79, 149.70, 143.36, 131.07, 130.99, 130.28, 129.55, 126.46, 123.77, 123.48, 121.84, 120.14, 119.91, 114.57, 114.42, 107.48, 106.75, 60.61, 60.53, 56.58, 56.14, 25.32. HRMS *m*/*z* calcd. for C_29_H_29_N_2_O_4_, [M + H]^+^: 469.2122, found: 469.213.

*(E)-1-(4-(ethyl(2-methylquinolin-4-yl)amino)phenyl)-3-(3,4,5-trimethoxyphenyl)prop-2-en-1-one* (**13f**). Yellow powder, Yield, 62%, m.p.148–150 °C. ^1^H NMR (400 MHz, DMSO-*d*_6_) δ 8.01 (dd, *J* = 8.3, 6.4 Hz, 2H), 7.85–7.70 (m, 2H), 7.63–7.57 (m, 1H), 7.55–7.46 (m, 1H), 7.45 (d, *J* = 8.6 Hz, 1H), 7.17 (s, 1H), 6.84–6.76 (m, 1H), 6.68 (d, *J* = 8.8 Hz, 1H), 6.63–6.57 (m, 1H), 4.00 (q, *J* = 7.0 Hz, 1H), 3.94 (q, *J* = 7.0 Hz, 1H), 3.84 (s, 3H), 3.70 (s, 2H), 3.63 (d, *J* = 5.6 Hz, 3H), 3.41 (d, *J* = 5.2 Hz, 3H), 2.70 (d, *J* = 7.4 Hz, 3H), 1.23 (dt, *J* = 21.2, 7.0 Hz, 3H). ^13^C NMR (100 MHz, DMSO-*d*_6_) δ 186.97, 160.87, 153.57, 151.83, 150.24, 149.91, 143.17, 139.92, 131.16, 131.01, 130.21, 129.64, 128.09, 126.53, 124.09, 123.73, 121.89, 121.14, 114.07, 106.73, 60.60, 56.57, 47.26, 25.38, 13.32. HRMS *m*/*z* calcd. for C_30_H_31_N_2_O_4_, [M + H]^+^: 483.2278, found: 483.2290.

### 3.3. Cell Culture

Cell lines used were cultured in humidified incubator at 37 °C and 5% CO_2_. The RPMI-1640 medium was supplemented with 10% fetal bovine serum, penicillin (100 U/mL), and streptomycin (0.1 mg/mL). All the cells were obtained from the Cell Bank of Type Culture Collection of Chinese Academy of Sciences (Shanghai, China).

### 3.4. MTT Assay

Cell lines were seeded into 126-well plates and incubated for 24 h. Then, cells were treated with different concentrations of compounds. Additionally, after another 48 h, MTT reagent (20 μL per well) was added and then incubated at 37 °C for 4 h. Formazan was then dissolved with DMSO. Absorbencies of formazan solution were measured at 4120 nm. The IC_50_ values of tested compounds were calculated by SPSS version 17.0.

### 3.5. Western Blotting Analysis

Gastric cancer cells were seeded in dishes and treated with **12e** or DMSO. After 48 h, MGC-803 cells were collected and then lysed. The denatured lysates of each group were electrophoretic separated in SDS-PAGE. Proteins were then transferred onto PVDF membranes from gels. After blocking for 2 h, membranes were incubated with primary antibodies conjugation. Then, the membranes were washed and incubated with 2nd antibodies. At last, specific proteins were detected.

### 3.6. General Methods

In this work, some other assays including colony formation assay were referred to our previous work [30,31].

## 4. Conclusions

Chalcone and quinoline are common scaffolds found in many compounds with many versatile pharmacological activities, especially anticancer activities. In this work, we assumed that linking chalcone fragment to the quinoline scaffold through the principle of molecular hybridization strategy could produce novel compounds with potential anticancer activity. Therefore, quinoline-chalcone derivatives were designed and synthesized, and we explored their antiproliferative activity against MGC-803, HCT-116, and MCF-7 cells. Among these compounds, compound **12e** exhibited a most excellent inhibitory potency against MGC-803, HCT-116, and MCF-7 cells, with IC_50_ values of 1.38, 5.34, and 5.21 µM, respectively, which were much lower than that of **5-FU** (IC_50_ values = 6.22, 10.4, and 11.1 μM, respectively). Further mechanism studies suggested that compound **12e** inhibited the cell colony formation activity of MGC-803 cells in a dose-dependent manner. Meanwhile, compound **12e** could arrest MGC-803 cells at the G2/M phase and significantly upregulate levels of apoptosis-related proteins (Caspase3/9 and cleaved-PARP) in MGC-803 cells. In addition, compound **12e** could significantly induce ROS generation, and was dependent on ROS production to exert inhibitory effects on gastric cancer cells. Taken together, all the results suggested that compound **12e** might be a valuable lead compound for the development of anticancer agents.

## Data Availability

Data of the compounds is available from the authors.

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
