# Peer review of "Design, Synthesis, and Anticancer Activity Studies of Novel Quinoline-Chalcone Derivatives"

_molecules, 2021, doi:10.3390/molecules26164899_

Round 1
Reviewer 1 Report
Dear Authors,
The manuscript ID: molecules-1299243 entitled “Design, Synthesis and Anticancer Activity Studies of Novel Quinoline-Chalcone Derivatives” written by Yong-Feng Guan, Xin-Ying Yuan, Wen-Bo Liu, Yin-Ru Li, Guang-Xi Yu, Xin-Yi Tian, Yan-Bing Zhang, Jian Song, Wen Li and Sai-Yang Zhang is devoted to new compounds with potential anticancer activity.
The paper presents interesting, original and innovative research and results. The development of cancer registries throughout the world has led to a search for novel anticancer drugs. I agree with the Authors that the molecular hybridization strategy is extensively used in drug design and discovery based on the combination of different bioactive moieties to produce new hybrids with the improved activities (often of a synergistic effect). Design and synthesis of novel quinoline-chalcone derivatives is an interesting solution in the fight against cancer.
This manuscript (Introduction, Results and Discussion, Materials and Methods, Conclusions) is properly written and organized. The title of the manuscript is accurate and express the main idea of manuscript. The Abstract clearly summarize the article contents. Introduction contains general data on chalcones and quinolines as anticancer agents. The purpose of the work is concise and concrete. Appropriate methods are used to perform these studies. Results are documented and right interpreted. Based on the results, adequate conclusions were drawn that one of the compounds (compound 12e) might be a valuable lead compound for the development of anticancer agents.
I have some suggestions in order to improve paper, which are the following:
- Please standardize the references in the text in accordance with the instructions for authors;
- Introduction: „used in in drug design” – „used in drug design”
3) 2. Results and Discussion
2.1. Chemistry – please standardize the font
4) Please standardize upper / lower case letters in subsection titles
I think that this article is valuable and worth publishing in Molecules, after minor review.
With highest regards,
Author Response
. Please standardize the references in the text in accordance with the instructions for authors;
Response: Thanks for your valuable comments. We feel sorry for our carelessness and have corrected it according to your valuable comments in the revised manuscript.
2. Introduction: „used in in drug design” – „used in drug design”; 3) 2. Results and Discussion
Response: Thanks for your valuable comments. We feel sorry for our carelessness and have corrected it according to your valuable comments in the revised manuscript.
3. Chemistry – please standardize the font; Please standardize upper / lower case letters in subsection titles
Response: Thanks for your valuable comments. We feel sorry for our carelessness and have corrected it according to your valuable comments in the revised manuscript.
Reviewer 2 Report
12 a-13 f are misplaced in figure 3.
are presented in the supplementary materials
13a,13b, 13e,13f in scheme 1
R2 before R1
5-Fu what is it?
3.2 12 a-12m
12k and 12k are different comppounds?
No information for 12 l,m
no discussion section is presented.

Author Response
- 12 a-13 f are misplaced in figure 3.are presented in the supplementary materials;
13a,13b, 13e,13f in scheme 1 R2 before R1
Response: Thanks for your valuable comments. We feel sorry for our carelessness and have corrected it according to your valuable comments in the revised manuscript.
2.5-Fu what is it?
Response: Thanks for your valuable comments. 5-Fluorouracil, abbreviated as 5-FU, is the most widely used anti pyrimidine drug in clinic with good curative effect on gastrointestinal cancers and other solid tumors. 5-FU is also often used as a positive control drug in many researches. We feel sorry for our carelessness and have added it according to your valuable comments in the revised manuscript.
- 12 a-12m 12k and 12k are different comppounds? No information for 12 l, m
no discussion section is presented.
Response: Thanks for your valuable comments. We feel sorry for our carelessness and have corrected it according to your valuable comments in the revised manuscript.
Reviewer 3 Report
Some suggestions:
In the introduction, I suggest to talk a bit more about Chalcone and quinoline by using these references:
- Anti-cancer chalcones: Structural and molecular target perspectives
European Journal of Medicinal Chemistry
Volume 98, 15 June 2015, Pages 69-114
https://doi.org/10.1016/j.ejmech.2015.05.004
- Quinoline as a privileged scaffold in cancer drug discovery
DOI: 10.2174/092986711795328382
In the experimental work, I suggest to cite the solvent system used in the flash chromatography with Rf if it is possible.
In the supplementary doc.:
Products (12b, 12e, 12i, 12g): HMRS doesn’t show enough purity of the compound, replace the existing HMRS by purer samples.
Writing errors:
- Via (should be written in italic)
- Figure 3. Authors used the wrong arrow (it is the retrosynthetic arrow , they should use the synthetic arrow
- In the supplementary doc.: HMRS not HNMRS for the compound 12e.
References: The authors follow the journal guidelines (Abbreviated Journal Name Year, Volume, page range) but they should make sure that the volume is written in italic.
Author Response
- In the introduction, I suggest to talk a bit more about Chalcone and quinoline by using these references:
Response: Thanks for your valuable comments. We feel sorry for our carelessness and have added it according to your valuable comments and cited these references in the revised manuscript.
- In the experimental work, I suggest to cite the solvent system used in the flash chromatography with Rf if it is possible.
Response: Thanks for your valuable comments. Due to the limitation of experimental conditions, we could not cite the solvent system used in the flash chromatography with Rf.
- In the supplementary doc.
Response: Thanks for your valuable comments. We feel sorry for our carelessness and have replaced it according to your valuable comments in the revised manuscript.
- 4. Writing errors: Via (should be written in italic) Figure 3. Authors used the wrong arrow (it is the retrosynthetic arrow, they should use the synthetic arrow In the supplementary doc.: HMRS not HNMRS for the compound 12e.
Response: Thanks for your valuable comments. We feel sorry for our carelessness and have corrected it according to your valuable comments in the revised manuscript.
Reviewer 4 Report
This manuscript written by Yong-Feng Guan and coworkers described the synthesis and characterization of some original quinoline-chalcone derivatives. The article is interesting and the authors did a good job, and I think this can be published in Molecules after some revision.
First, the authors did not discuss anything about spectral interpretation. The spectral data are listed, without an exact assignment.
It would also be good to know where the authors came to the conclusion that the E isomer results from the synthesis? The geometric isomerism of the compounds must also be discussed through spectra.
IR absorption spectra would also be very useful.
In Figure 4 the authors considered 4-chloro-substitution twice. What is the correct order of intensity of the biological effect?
To the Material and Methods must be added the manufacturers and the country of the manufacturers of the equipments used in the analysis.
Chapters 3.1. and 3.2 contain the most inadvertent. These must be rewritten in correlation with Scheme 1. The numbering of the compounds is wrong in the text and the working methodology is not understandable
As for the chemical names of the compounds, they must be written without spaces; you must use the space before the parentheses with the numbering of the compounds. It would be great to review these chemical names
Author Response
- It would also be good to know where the authors came to the conclusion that the E isomer results from the synthesis? The geometric isomerism of the compounds must also be discussed through spectra.
Response: Thanks for your valuable comments. It is widely reported in many literatures that chalcone produced by aldehyde ketone condensation reaction under the condition of sodium hydroxide and ethanol is E isome.
- In Figure 4 the authors considered 4-chloro-substitution twice. What is the correct order of intensity of the biological effect?
Response: Thanks for your valuable comments. We feel sorry for our carelessness and have corrected it according to your valuable comments in the revised manuscript.
- To the Material and Methods must be added the manufacturers and the country of the manufacturers of the equipments used in the analysis.
Response: Thanks for your valuable comments. We feel sorry for our carelessness and have added it according to your valuable comments in the revised manuscript.
- Chapters 3.1. and 3.2 contain the most inadvertent. These must be rewritten in correlation with Scheme 1. The numbering of the compounds is wrong in the text and the working methodology is not understandable
Response: Thanks for your valuable comments. We feel sorry for our carelessness and have corrected it according to your valuable comments in the revised manuscript.
- As for the chemical names of the compounds, they must be written without spaces; you must use the space before the parentheses with the numbering of the compounds. It would be great to review these chemical names Suggested "for cancer therapy and drug resistance"
Response: Thanks for your valuable comments. We feel sorry for our carelessness and have corrected it according to your valuable comments in the revised manuscript.
Round 2
Reviewer 4 Report
I consider that the authors found my observations justified and I found that a lot of improvements were made to the article so that it could be published.